# Effects of Polyvinyl Chloride (PVC) Microplastic Particles on Gut Microbiota Composition and Health Status in Rabbit Livestock

**DOI:** 10.3390/ijms252312646

**Published:** 2024-11-25

**Authors:** Péter P. Papp, Orsolya Ivett Hoffmann, Balázs Libisch, Tibor Keresztény, Annamária Gerőcs, Katalin Posta, László Hiripi, Anna Hegyi, Elen Gócza, Zsuzsanna Szőke, Ferenc Olasz

**Affiliations:** 1Agribiotechnology and Precision Breeding for Food Security National Laboratory, Department of Microbiology and Applied Biotechnology, Institute of Genetics and Biotechnology, Hungarian University of Agriculture and Life Sciences, 2100 Gödöllő, Hungary; pppeter507@gmail.com (P.P.P.); libisch.balazs.karoly@uni-mate.hu (B.L.); kereszteny.tibor@uni-mate.hu (T.K.); ancsi.g@gmail.com (A.G.); posta.katalin@uni-mate.hu (K.P.); hegyian@gmail.com (A.H.); 2Agribiotechnology and Precision Breeding for Food Security National Laboratory, Department of Animal Biotechnology, Institute of Genetics and Biotechnology, Hungarian University of Agriculture and Life Sciences, 2100 Gödöllő, Hungary; hoffmann.orsolya.ivett@uni-mate.hu (O.I.H.); or hiripi.laszlo@semmelweis.hu (L.H.); gocza.elen@uni-mate.hu (E.G.); 3Doctoral School of Biological Sciences, Hungarian University of Agriculture and Life Sciences, 2100 Gödöllő, Hungary; 4Doctoral School of Biology, ELTE Eötvös Loránd University, 1117 Budapest, Hungary; 5Laboratory Animal Science Coordination Center, Semmelweis University, Nagyvárad tér 4, 1089 Budapest, Hungary

**Keywords:** microplastic, microbiota, rabbit, toxic effect, hormone level, physiological status, polyvinyl chloride (PVC)

## Abstract

The widespread use of polyvinyl chloride (PVC) and its entry into humans and livestock is of serious concern. In our study, we investigated the impact of PVC treatments on physiological, pathological, hormonal, and microbiota changes in female rabbits. Trend-like alterations in weight were observed in the spleen, liver, and kidney in both low (P1) and high dose (P2) PVC treatment groups. Histopathological examination revealed exfoliation of the intestinal mucosa in the treated groups compared to the control, and microplastic particles were penetrated and embedded in the spleen. Furthermore, both P1 and P2 showed increased 17-beta-estradiol (E2) hormone levels, indicating early sexual maturation. Moreover, the elevated tumor necrosis factor alpha (TNF-α) levels suggest inflammatory reactions associated with PVC treatment. Genus-level analyses of the gut microbiota in group P2 showed several genera with increased or decreased abundance. In conclusion, significant or trend-like correlations were demonstrated between the PVC content of feed and physiological, pathological, and microbiota parameters. To our knowledge, this is the first study to investigate the broad-spectrum effects of PVC microplastic exposure in rabbits. These results highlight the potential health risks associated with PVC microplastic exposure, warranting further investigations in both animals and humans.

## 1. Introduction

Microplastic pollution has emerged as a significant environmental concern, impacting both ecosystems and human health [1,2,3,4,5]. The degradation of plastics, previously considered to be non-degradable or very difficult to degrade, produces microplastic particles (60–250 µm in size) small enough to cross biological barriers and penetrate living organisms, affecting their physiological state [6,7]. Therefore, it seems necessary to introduce regulations to mitigate microplastic pollution and its harmful effects. Currently, there is no common regulation in the European Union regarding microplastic threshold limit values. However, the European Commission published a proposal in 2022 to restrict intentionally added microplastics, which could ban the use of microplastics in cosmetics, cleaning products, and pesticides [8].

Microplastics can enter animals through the food chain and from environmental sources (air, water) [9,10,11]. Penetration occurs mainly in the intestinal tract, but after breaking the barrier, microparticles can also be present in various organs and fluids of the animals [12,13]. Initial research has predominantly focused on fish and other marine organisms [14,15,16,17]. In contrast, limited data are currently available for terrestrial mammals other than humans, mice, and rats [18,19,20,21,22]. Studies have also been conducted in mammalian farm animals [23,24,25]. However, very few studies have been published on microplastic exposure in rabbits and its effects on their physiological status [26,27], which underlines the timeliness and importance of the current study.

Microplastics have a wide range of potential effects, such as malnutrition, gastrointestinal inflammation, mechanical injury, dysbiosis, oxidative stress, and intestinal flora disruption. Additionally, their degradation into nanoplastics allows the penetration of biological barriers. Furthermore, microplastics may act as vectors for potential pathogens and foreign, non-resident bacteria. One of their most important effects is that they can accumulate in animals and humans, exerting toxic effects [28]. For example, microplastics have been suggested to be potentially responsible for inflammation and human lung cancer [29]. The effects of microplastic exposure are diverse, impacting metabolic pathways such as energy and nucleic acid metabolism [30,31], which are associated with gene expression changes, for example, genes related to glycolysis and lipid metabolism [17,31]. Microplastics and their additives have been shown to affect sex hormone levels and endocrine function in various organisms. Studies have revealed that exposure to microplastics and their associated chemical additives can lead to endocrine disruption and reproductive health issues [32,33,34,35]. It is also noteworthy that the prevalence of pathogens in the gut and in the body is related to microplastic load [36,37,38]. Furthermore, microplastics can also enhance the spread of antibiotic-resistance genes [39,40]. Polyvinyl chloride (PVC) microplastic particles and their degradation products are also present in our environment, affecting the physiological state of organisms exposed to pollution [41,42,43,44].

PVC was long considered a harmless material, although studies as early as 70 years ago, as summarized by Soffritti et al. [45], highlighted the harmful effects of PVC and PVC monomers on health. PVC is versatile in many industries, such as construction, automotive, pipe and cable manufacturing, and household goods, as it is durable, lightweight, and versatile. Despite this, PVC is considered one of the most dangerous plastics, not the plastic itself, but its potentially toxic degradation products, such as dioxin and other chloride-containing toxic degradation products. Recently, several studies have been published on the health effects of PVC [46,47,48]. Studies describe the possible harmful effects of PVC dust, vinyl chloride monomers, and toxic substances released from water pipes and entering the water by cleaning and disinfecting agents. These potentially harmful effects can be very diverse, including brain cancer, neurological effects, and pulmonary carcinogenesis through persistent alveolar inflammation, alveolar macrophage activation, developing liver angiosarcoma, and release of growth factors [49,50].

The gut flora of healthy animals is optimal, a condition known as eubiosis, which is essential for maintaining health, feed conversion, and production efficiency. The gut microbiota plays an important role in maintaining the eubiotic state in animal health [20,51]. Based on studies in mice, polystyrene treatment caused 15 significant changes at the genus level, with significant alterations in the metabolic pathways of the bacterial community, and caused intestinal microbial dysbiosis and liver disease [18,52]. Another study in mice showed similar results: when examining phylum-level changes, the proportion of Firmicutes (recently Bacillota), Bacteroidetes (recently Bacteroidota), and α-Proteobacteria (recently Pseudomonadota) in the microbiota decreased [53]. These experiments have shown that microplastic treatment affects the microbial composition of the gastrointestinal tract [54,55]. Thus, the disbalance of the microbiota leads to a dysbiotic state, which stresses the immune system of the animals, reduces the efficiency of feed conversion, decreases productivity, and in severe cases, disease and mortality can occur [55,56,57,58,59].

In the present work, we demonstrate the effects of PVC exposure in an in vivo animal model [60,61,62,63]. Our study highlights the multiple effects of PVC microplastic exposure on rabbits. Significant histological and pathological changes, putative reproductive effects, and significant alterations of the gut microbiota were demonstrated. These results justify the need for further research on this emerging topic.

## 2. Results

### 2.1. Determination of the PVC and Mycotoxin Content of the Feed Used in the Experiments

Before starting the in vivo experiment, it had to be ensured that no microplastics were introduced into the animals from the control feed or drinking water during the experiments, as their presence could potentially distort the experimental results. The results confirmed that PVC microplastic particles were only detected in the feed of the P1 (low dose) and P2 (high dose) treatment groups (Appendix A). No microplastic contamination was found in the drinking water.

Mycoestrogens (zearalenone and its different metabolites) may have an effect on early sexual maturation [64,65] and may affect the gut microbiome [66], so it was necessary to determine the mycotoxin content of the feed. FB1-toxin was not detected in the diet, but DON, aflatoxin, and ZEN were present (Appendix A): DON: 182.16 to 250.68 ng/g; aflatoxin: 11.64 to 12.05 ng/g; ZEN: 19.0 to 24.9 ng/g. However, these values were within the permitted limits (Commission Recommendation 2006/576/EC). There was no significant difference in the toxin levels between the diets fed to the C, P1, and P2 treatment groups.

### 2.2. Changes in Body Weight During the Feeding Experiment

The change in body weight of animals can be indicative of the general health status, as insufficient weight gain may indicate disease or poisoning. Body weight measurements were taken at one-week intervals in all three treatment groups for 35 days starting with the pre-feeding period. In all three treatment groups, the initial mean body weights of the rabbits were nearly identical (3704–3732 g) (Appendix A). The average body weight of each treatment group at the end of the feeding trial was 4355 g (C), 4370 g (P1), and 4259 g (P2). In some cases, both weight gain and weight loss were observed in certain animals at specific time points. Overall, as the mean body weights were very similar both at the start and at the end, no significant differences could be observed between treatment groups.

### 2.3. Analysis of Organ Weight After PVC Treatment

Changes in the organ weights can be a consequence of the effects of PVC exposure. In the physiological study of the experimental animals, the weight of the spleen, kidney, liver, and ovary was measured for each animal in each treatment group (C, P1, and P2) (Figure 1 and Appendix A). One-way ANOVA analysis showed that there were no significant differences in the weight changes of the spleen, kidney, liver, and ovary between the treatment groups. A trend-like decrease in the weight change of the liver and kidney was detected in the P1 and P2 groups compared to the control. The spleen showed a non-significant increase in weight after low and high PVC treatment compared to the control. No significant change in ovarian weight was detected.

### 2.4. Macroscopic Pathological Alterations of Organs

The examination of the morphological pathological lesions in the organs of the animals was conducted immediately after dissection (Appendix A). Photographic documentation of the state of the various organs was recorded (Appendix A). Our investigation primarily focused on the examination of the liver, kidney, abdominal lymph nodes, small intestine, caecum, colon, rectum, and spleen. The results showed no macroscopic abnormalities in the heart, lungs, feces, colon, and rectum.

Table 1 shows the number of organs with macroscopic abnormalities in relation to the total number of organs examined. The liver was found to be slightly swollen in one animal in the control group, three in the P1 group, and four in the P2 treatment group. The kidneys were healthy in all treatment groups, except for one rabbit in the control group whose kidney was slightly swollen. The Peyer’s plaques and lymph nodes were slightly swollen in all rabbits in the control group, four in P1, and two in P2. The mucosa of the small intestine was slightly swollen in two individuals in the control group, while in P1 and P2, this phenomenon was observed in all 6-6 individuals examined. In the caecum, the appendix extension was slightly enlarged in one individual of group P1, while in another individual it was flaccid. In three individuals of the control group, the appendix extension veins were slightly dilated. The spleen was slightly swollen in three individuals of the control group, while in treatment groups P1 and P2, swelling was detected in 5-5 cases.

### 2.5. Histological Alterations of the Organs

The macroscopic examination of some organs (liver, spleen, intestine) showed trend-like abnormalities, but these abnormalities were not significant except for the small intestine. Additional information on organ dysfunction was obtained by histological examination, which revealed focal or multifocal exfoliation of the ileal mucosa in groups P1 and P2 compared to the control animals (Figure 2). Exfoliation mainly affected the apical part of the intestinal villi in the P1 and P2 groups, and moderately increased lympho-histiocytic infiltration was also observed. No exfoliation or increased lympho-histiocytic infiltration was detected in the caecum and rectum, indicating that exfoliation affected only the ileum within the entire intestinal tract.

In the spleen of the P1 and P2 groups, congestion, sinusoidal dilation, and swelling of the macrophages were observed compared to the control animals. It is also noteworthy that in the spleen samples from rabbits in groups P1 and P2, phagocytosed plastic granules were visible in the swollen macrophages (Figure 2).

No histological alterations in the kidneys could be detected. In the liver, a “foamy” appearance of hepatocytes was observed in both the control and treated rabbits, causing steatosis, presumably related to the high (physiological) glycogen content. This kind of pathological alteration has also been observed in fishes after nano- and microplastic treatment [7,67].

### 2.6. Effect of PVC Treatment on Hormone Levels

We investigated the levels of the estradiol (E2) and progesterone (P4) hormones measured in the blood serum of female rabbits. Measurements were taken at one-week intervals, starting on the first day of the microplastic feedings. Neither significant nor trend-like changes could be detected in progesterone levels (P4), as there were very large differences between animals and even between different samples from the same animal.

Trend-like changes (Appendix A) could be detected in E2 levels as a result of the microplastic treatment. In the control animals, the estradiol (E2) levels decreased slightly from the starting value of 13.0 pg/mL to 9.9 pg/mL by week 3 at the end of the experiment. In the P1 animals, its value reached 39.2 pg/mL by week 3, starting from a baseline of 12.9 pg/mL. A similar trend was observed for the P2 group, where it rose from a baseline of 15.5 pg/mL to 44.7 pg/mL by week 3. The Kruskal-Wallis test revealed significant differences in E2 levels between treatment groups, with Dunn-Bonferroni tests showing significant increases in E2 levels for both P1 and P2 treatment groups compared to their respective baseline levels and the control group.

A similar trend-like effect was found when the number of mature follicles in the ovaries of female rabbits was examined. The mean number of mature follicles in the control animals was 2.25 ± 1.7, while the mean number of mature follicles in the P1 and P2 treatment groups was 4.0 ± 1.4 and 5.4 ± 2.4, respectively, at the end of the experimental period.

### 2.7. Determination of the Tumor Necrosis Factor Alpha Level

Tumor necrosis factor alpha (TNF-α) levels were measured in the blood serum (Appendix A). Measurements were performed at one-week intervals, starting at the beginning of the microplastic-feeding period.

In the control rabbits, the initial value was 8.5 pg/mL, which did not change significantly thereafter, and was 9.9 pg/mL at the end of sampling. In rabbits fed P1 microplastic, the value increased from 8.0 pg/mL to 41.3 pg/mL at the end. However, this value was mainly due to an outlier value measured in one animal (ID number 450: 141 pg/mL, Appendix A), which may indicate extensive inflammation. A similar trend was observed for the P2 group, where from a starting value of 4.0 pg/mL, it reached 26.6 pg/mL by the end of the sampling, which is different from the value observed in the control rabbits. It is noteworthy that while in the control group, only a few cases of increase in TNF-α level from baseline by the end of the experiment were observed, a trend-like continuous increase in TNF-α level was detected in the P2 group. The Bonferroni Post hoc test showed that the pairwise group comparison of P2_week1–P2_week3 has a *p*-value of less than 0.05 (*p* = 0.024), and thus, based on the available data, it can be assumed that the TNF-α levels in the two groups are significantly different.

### 2.8. Analyses of Blood Comprehensive Metabolic Panel

The concentration of the blood serum determinants was measured as described in the Material and Methods section. No differences were observed in any of the treatment groups for the following parameters: total protein, aspartate aminotransferase, α-amylase, glucose, creatinine, inorganic phosphorus, potassium, sodium, and chloride (Appendix A). However, for both microplastic-treated and control groups, values higher or lower than the reference were obtained for albumin, alanine aminotransferase, triglycerides, total cholesterol, and ionized calcium. For some parameters, such as urea, creatinine, and alkaline phosphatase, there were only one or two instances of deviations from the reference value. Additionally, due to high standard deviations, no significant difference or discernible trends were identified, except for alkaline phosphatase. For alkaline phosphatase, the microplastic-treated groups exhibited lower values compared to the reference, while the control group reached the reference value.

### 2.9. Alterations in the Diversity and Bacterial Composition of the Gut Microbiota in Response to PVC Microplastic Supplemented Diet

Amplicon sequencing of the 16S V3–V4 variable regions of the 16S rRNA gene yielded 95,000–130,000 merged and quality-filtered Illumina reads per intestinal sample, containing no ambiguous nucleotides. Amplicon sequencing data of the V3–V4 variable regions of the 16S rRNA gene was analysed by QIIME as microbial marker-gene sequence data to generate the gut bacterial taxonomic profiles for the examined intestinal segments and for community diversity analyses (see Section 4.9). The mean Illumina Quality Score in the entire 16S V3–V4 amplicon range was >Q30, and thus suitable for further analyses. For differential abundance testing, Kruskal–Wallis tests were performed. Only *p* < 0.05 values adjusted by the Bonferroni correction were considered statistically significant.

The α-diversity according to the Shannon index (Figure 3A) in the ileum, caecum, and stool samples did not show a significant difference between the three treatment groups, however, the mean value tended to be lower in the caecum and faeces of the P1 and P2 groups compared to the control group. On the other hand, the Chao1 index showed a significant decrease between the control and P1 treatment groups in the faeces, and between the control and the P2 treatment groups in the caecum (*p* < 0.05, Figure 3B).

The phylum- and family-level mean bacterial compositions of the ileum, caecum, and faeces in the three treatment groups are shown in Appendix A. The main phyla were the Firmicutes (recently Bacillota), Bacteroidetes (recently Bacteroidota), Verrucomicrobia, and Actinobacteria, while the dominant families included the *Ruminococcaceae*, *Lachnospiraceae*, *Eubacteriaceae*, *Verrucomicrobiaceae*, *Clostridiales vadin BB60 group*, *Christensenellaceae*, and *Rikenellaceae*. According to phylum-level differential abundance testing, the relative abundance of the Firmicutes and Bacteroidetes phyla did not show a significant change between the treatment groups, with the exception of a significantly increased Bacteroidetes in the P2 caecum compared to the control (*p* = 0.012, Appendix A). Further, the phylum Proteobacteria (recently Pseudomonadota) had a higher abundance in the P2 faeces compared to the control group (*p* = 0.045, Appendix A).

The genus-level comparison of the gut bacterial composition revealed a significant difference between the control and the high PVC-treated P2 group for the relative abundance of several genera, including an increase of *Odoribacter* and *Alistipes* in both the caecum and the faeces of the P2 group, an increase of *Rikenella* and *Phascolarctobacterium* in the faeces of the P2 group, and a decrease of a *Clostridiales vadin BB60 group* uncultured bacterium and the *Clostridiales FamilyXIII AD3011 group* in the P2 faeces. Moreover, *Fusicatenibacter* and *Oxalobacter* decreased significantly in the caecum between the control and the high PVC group, and the *Ruminococcaceae UCG-005 group* increased significantly in the caecum between the control and the P2 groups (Figure 4 and Figure 5, Appendix A).

Comparing P1 and P2 abundance values, significant differences were observed (Appendix A). The *Flavobacteriaceae uncultured* genus showed an increase in the P2 group compared to the P1 group when comparing their abundance, as the *p*-value for both caecum, ileum, and faeces showed a significant difference (*p* < 0.05). However, a decrease was detected for *Eggerthella* in both P2 ileum and P2 faeces, and for *Ruminococcaceae UCG-007* and *Ruminococcaceae NK4A214 group* in P2 faeces (Appendix A).

Overall, the relative abundances of several genera showed significant differences in a similar pattern across multiple intestinal segments (Appendix A). These findings indicate that changes in the bacterial composition of the examined intestinal sections were correlated with each other in response to PVC microplastic feeding for several genera.

Significant correlations were detected between the TNF-α values in the blood serum at week 3 (Appendix A), swollen liver (Table 1), or the PVC microplastic content of the experimental rabbit feeds, and between the relative abundances of certain gut bacteria (Table 2). The phyla Saccharibacteria and Cyanobacteria both had non-significantly (*p* > 0.05) elevated levels in the P2 group caecum and faeces (Appendix A).

## 3. Discussion

In our study, we investigated the effects of PVC treatment on physiological, pathological, hormonal, and microbiota changes in rabbits. Although there are some publications on the microplastic exposure of rabbits and its effects on their health, this topic is understudied and there is no comprehensive publication with this scope. As very few publications are available in this area, comparisons are often based on experiments in mice or rats, knowing that these comparisons unfortunately only allow tentative conclusions, as the intestinal digestion of rabbits and rodents differs significantly, since in rabbits the caecum plays an important role, unlike in rodents. Given the widespread use of PVC and its potential to enter human and animal bodies via drinking water, feed, and inhalation, it is timely to further investigate these issues. To provide further reliability to our experiments, the rabbit feed and drinking water used in the experiments were tested for microplastic and mycotoxin contamination. Based on the negative results obtained, it was established that external conditions did not influence the results of the experiments.

Microplastics have been shown to have potential effects on obesity in mice. Interestingly, research has shown that microplastics combined with antibiotics can disrupt gut microbiota homeostasis in mice, leading to brain lesions and inflammation through the gut–brain axis [68]. This finding is particularly relevant as gut microbiota dysbiosis has been associated with obesity and metabolic disorders in both rodents and humans. Furthermore, microplastics have been detected in the human placenta and blood, raising concerns about their potential long-term effects on human health, including possible impacts on metabolism and obesity [69]. In conclusion, while there is evidence suggesting that microplastics can cause inflammatory responses and disrupt gut microbiota in mice, which are factors associated with obesity, direct evidence linking microplastics to obesity in humans is limited. Further research is needed to fully understand the potential relationship between microplastic exposure and obesity in humans.

Trend-like changes were observed in the spleen weight, where the spleen weight increased in the P1 and P2 groups, while the average weight of the liver and kidney decreased. A similar observation was made of the liver in mouse offspring after polystyrene treatment, as the liver mass decreased. In contrast, an increase in liver weight was observed by Zhao et al. [70] in mice. In our case, the trend-like change in liver weight is consistent with the observed data on macroscopic organ alterations, as abnormalities were observed in 3/6 cases in P1 and 4/6 cases in P2, whereas only in 1/6 cases in the control group showed macroscopic alterations.

The decrease in kidney mass is consistent with previous observations on polystyrene-treated mice [71], but in this study, this decrease was not manifested by macroscopic and histological abnormalities. Regarding the spleen, although both the P1 and P2 experimental groups showed a higher number of macroscopic abnormalities (5/6) than the control (3/6), no clear conclusion could be drawn. However, the histological examination of the spleen indicated sinusoidal dilation and swelling of the macrophages. Macrophages, as part of the immune system, are affected by ingested microplastic particles with a broad range of effects [72,73]. Moreover, the spleen samples from groups P1 and P2 contained phagocytosed plastic granules in the swollen macrophages. This finding is in line with previous publications describing the entry of microplastic particles into animal bodies [6,7,73,74]. However, no data on the entry of PVC by phagocytosis into mammalian bodies have so far been observed. Thus, we provide the first evidence that PVC is able to enter mammals, in our case rabbits, by phagocytosis of macrophages.

PVC treatment caused very noticeable lesions in the intestine, where significant macroscopic lesions such as swollen mucosa were detected in the ileum. Notably, PVC content in the diet was positively correlated with small intestine swollen mucosa lesions (R = 0.506, *p* < 0.05). This finding was also confirmed by histological examination because multifocal exfoliation of the ileal mucosa and damaged villi were detected in animals belonging to groups P1 and P2 compared to control animals. This result is consistent with the observation, where damaged villi were observed in broiler chickens after chlorogenic acid exposure [75] and in fish after PVC treatment [76].

Microplastics and/or their degradation products may also affect the reproductive biology of humans and animals [32,33]. Interestingly, the effects of microplastics on sex hormones can be sex specific. For example, male *Drosophila* flies have been found to be generally more susceptible to the toxic effects of polystyrene microplastics than female flies, showing greater mortality and metabolic disruptions [34]. This suggests that the impact of microplastics on sex hormone levels may vary between males and females, highlighting the complexity of their effects on endocrine function. In conclusion, while the exact mechanisms are still being studied, there is growing evidence that microplastics and their additives can indeed affect sex hormone levels and endocrine function. In female animals, the ovaries produce large amounts of estrogens, of which 17-beta-oestradiol (E2) is the most important and biologically active product of the ovaries. A decreasing level of E2 indicates impaired sexual function. On the opposite, an increasing level of E2 can indicate early sexual maturation, as described by Leonardi et al., [77] for the effect of bisphenol A. In our experiments, the level of the estrogen hormone E2 in the control group stagnated compared to the baseline value. The period investigated (12–17 weeks of age) was still the initial stage of sexual maturation of the rabbits, as sexual maturation in NZW rabbits is 16–24 weeks [78], so at most a small increase in E2 hormone levels can be expected in the control. In contrast, in both the low (P1) and high (P2) dose-treated animals, higher E2 hormone levels were measured at the end of the experimental period compared to the initial value. Several compounds can affect reproduction [79] but the effects of PVC on hormones and the reproductive system have not been measured in mammals. However, we found a correlation with experiments by Yang et al. [80] in mice exposed to polystyrene microplastic treatment, where increased blood serum E2 levels were detected. The low concentration of the mycotoxin zearalenone found in feeds is well below the permitted value; this concentration could not have caused premature puberty in our experiment [64,81]. The early sexual maturation of female rabbits is also confirmed by the higher follicle counts measured in PVC-treated females compared to the control. In conclusion, while the exact mechanisms are still being studied, there is growing evidence that microplastics and their additives can indeed affect sex hormone levels and endocrine function.

Tumor necrosis factor alpha (TNF-α) is a pro-inflammatory cytokine that plays a central role in the regulation of the inflammatory response and is produced primarily by activated macrophages [82,83,84]. It has several functions, stimulating inflammatory processes, increasing fever, inducing apoptosis, and stimulating the so-called acute phase response. Busch et al., [85] demonstrated the micro- and nanoplastic inflammatory effects of PVC in a cell culture system. Thus, the measurement of its levels is indicative of the inflammatory processes taking place in the body. In P1 rabbits, a higher level was obtained than in the control, indicating a trend-like increase in TNF-α level. A similar trend was observed for the P2 animals, i.e., from a starting value of 4.0 pg/mL, it reached 26.6 pg/mL by the time of the end of sampling, which is significantly different from the value observed in control rabbits (9.9 pg/mL). These results suggest that PVC treatment induces inflammatory processes in treated rabbits, which is consistent with our results on pathological and histological alterations of the small intestine and correlates with the observation of Busch et al. [85] in cell culture.

Microplastic treatment can alter the gut microbiota, as it has been described in several publications [18,52,54,55,83,86]. Therefore, it was investigated how PVC microplastic treatment affects the composition of the microbiota, and several significant changes were found in both P1 and P2 PVC microplastic-treated rabbits compared to control animals.

The diversity of the gut microbiota is a good indicator of the health of the host [87,88]. According to our data, the Chao1 index showed a significant decrease in diversity between the control and treated groups P1 or P2 in the coecum and faecal samples (Figure 3B), and the Shannon index also showed a decreasing trend in the caecum and the faeces in the PVC groups (Figure 3A). These results may indicate that PVC microplastic treatment reduces the diversity of the gut microbiota, implying that the adaptability of the microbiota has been compromised.

Significant differences were detected by comparing genus-level abundance data of the control and high-dose PVC-treated rabbits. Increases were detected in *Odoribacter*, *Alistipes*, *Rikenella*, *Phascolarctobacterium*, and *Ruminococcaceae UCG-005 group*, while the *Clostridiales vadin BB60 group*, *Clostridiales FamilyXIII AD3011 group*, *Fusicatenibacter* and *Oxalobacter* showed significant decreases. We analysed the potential roles of these changes in the microbiota in response to stress caused by PVC treatment and compared them to data obtained from other studies.

Since we detected several significant or trend-like physiological, pathological, and microbiota abundance changes in response to PVC treatment, it was necessary to investigate whether there is a correlation between alterations in physiological and pathological parameters and microbiota changes. Our analysis showed significant correlations between the obtained TNF-α values (Appendix A) and the PVC microplastic contents of the experimental rabbit feeds and between the relative abundances of certain gut bacteria (Table 2). The phyla Saccharibacteria and Cyanobacteria both had non-significantly (*p* > 0.05) elevated levels in the P2 group caecum and faeces (Appendix A).

*Odoribacter* can cause health problems both in humans and in domestic animals. However, their presence has also been shown to be beneficial in preventing certain diseases [89]. In mice administered intranasally with nanoplastics, *Odoribacter* was among the most correlated genera associated with nanoplastics treatment in the nasal microbiota, and it was also suggested that it played an important role in maintaining the stability of the lung microbiota network in the microplastic-treated mice group [90]. In our case, it was also observed that the abundance of the *Odoribacter* genus correlated with the PVC content of the feed (Table 2).

The genus *Alistipes* has been shown to have a protective effect on a number of human diseases [91]. When the effects of nanoplastics (nano-polystyrene) on the intestinal health and growth performance of juvenile *Larimichthys crocea* were investigated, a significant increase in the potentially pathogenic *Alistipes* was found [92]. *Alistipes* also often increases in patients with intestinal stress syndrome [93]. *Alistipes* may affect the utilisation of tryptophan and disturb the 5-hydroxytryptamine (serotonin) system in the intestine, which is involved in the regulation of various physiological functions and pathological states [92,94]. It should be noted that an increase of *Alistipes* abundance in the control vs. P2 comparison was also detected (Figure 5A), and a significant correlation of its abundance with the rabbit feed PVC content (Table 2). Furthermore, the relative abundance of *Alistipes* in both the caecum (R = 0.479, *p* < 0.05) and faeces (R = 0.587, *p* < 0.01) was positively correlated with liver lesions (Table 1). The presence of the *Alistipes* genus in certain liver diseases was suggested to be linked with the healthy state [95], however, exposure to microplastics in several studies caused liver damage together with an associated increased *Alistipes* abundance in the gut microbiota [96,97,98], similar to the current study (Figure 5A, Table 2). Thus, liver damage caused by microplastic exposure may be associated with an increase in the abundance of *Alistipes* and other gut bacteria [98].

The *Rikenella* genus (see Figure 4 and Table 2) has been detected in the digestive tract of various animals, and its increased presence in mice has been associated with lupus and Alzheimer’s disease [99,100]. The increased levels of *Rikenella* genus in our case may also be a consequence of the stress caused by PVC exposure, and its abundance showed a significant correlation with the PVC content of the feed (Table 2).

The presence of certain stressors has increased the abundance of the *Phascolarctobacterium* genus (Figure 4 and Table 2) in several reports, such as sodium alginate and oxymatrine used to treat liver fibrosis [101], or microplastic exposure [102]. Similarly, simo decoction [103] used for the treatment of spleen and gastrointestinal diseases increases the abundance of the genus. In our current study, an increase of the opportunistic pathogen *Phascolarctobacterium* was found in the PVC-treated rabbit intestine, similar to its elevated level in PET microplastic-exposed human gut microbiota [104]. We also observed an abundance increase of *Phascolarctobacterium* in the PVC-treated rabbits (Table 2).

An increased abundance of *Rumminococcaceae UCG-005* was found in the PVC-treated rabbits in the current study, and a significant correlation of its relative abundance with TNF-α levels in the blood serum on week 3 (Figure 4, Table 2, Appendix A). The presence of *Rumminococcaceae UCG-005* has been reported primarily in human cases of disease [105]. However, loperamide-induced constipation in mice also resulted in an altered microbiota with a higher abundance of *Ruminococcaceae* UCG-005 [106]. In addition, recent studies reported that UCG-005 might exert an effect on the development of paclitaxel-induced peripheral neuropathy in rats [107] and may promote other painful manifestations in rodents [107,108]. *Ruminococcaceae* UCG-005 was positively correlated with diarrhoea incidence and negatively correlated with feed conversion ratio in the faeces of weaning piglets [109], and UCG-005 was also more abundant in the rabbit line of lower longevity and the rabbit group of lower number of parities in a study of the rabbit gut microbiome composition as a potential predictor of longevity in rabbits [88].

The *Clostridiales vadin BB60 group* was suggested to be associated with various health benefits, such as in the colorectum of layer chickens, where pathogenic *Clostridium* was negatively correlated with the *Clostridiales vadin BB60 group* [110]. It was proposed that the *vadin BB60 group* is likely to play an important role in establishing healthy caecal flora in broiler chickens, where *S.* Typhimurium infection caused a remarkable decrease in the abundance of the *Clostridiales vadin BB60 group* [111]. When chlorogenic acid exerted beneficial effects in broiler chickens, this was associated with an increased abundance of the short-chain fatty acid-producing bacteria of the *vadin BB60 group* [75]. In accordance with these earlier observations, under dietary PVC exposure, the *Clostridiales vadin BB60 group* decreased significantly (*p* < 0.01) in the faeces of both the P1 and P2 rabbit groups compared to control, indicating a negative effect of microplastic treatment on rabbit gut microbiota, suggesting that *Clostridiales vadin BB60 group* is likely to play a role in establishing healthy caecal flora in rabbits as well.

Similar to the *Clostridiales vadin BB60 group*, *Oxalobacter* also showed a decrease in the caecum between the control and the P2 group. The genus *Oxalobacter* is able to utilise oxalate as a carbon source and has been detected in the intestinal tract, where its abundance was also related to the actual state of the gut microbiome [112,113]. When rabbits were exposed to zearalenone treatment, the microbiota composition was significantly affected and *Oxalobacter* abundance decreased [114]. In our case, this may be indicative of a disruption of the gut microbiome in the PVC-treated rabbits, causing dysbiosis.

At the phylum level, Proteobacteria (recently Pseudomonadota) increased significantly in the faeces of the high-PVC rabbit group, and within this phylum, the Burkholderiales and Desulfovibrionales orders both had an increasing trend compared to controls (Appendix A). Likewise, exposure of mice to 2-μm PVC microplastics led to a higher level of Proteobacteria in the exposed group compared with the control [115], similar to the effect of 5-μm polystyrene microplastics that triggered a significant increase in Proteobacteria as well as other potentially pro-inflammatory bacteria, such as Desulfovibrio in mice gut microbiota [97]. Moreover, the increasing tendency of the phyla Cyanobacteria and Saccharibacteria in the P2 caecum and/or faeces positively correlated with the blood serum TNF-α values measured on week 3 (Table 2 and Appendix A). When freshwater crabs (*Eriocheir sinensis*) were exposed to four different microplastic concentrations, the highest concentration with a 21-day exposure resulted in dysbiosis of the gut microbiota, characterised by the increased relative abundances of the phyla Cyanobacteria, Proteobacteria, and other taxa [116]. Multiple studies have demonstrated the overrepresentation in the plastisphere of the phyla Proteobacteria and Cyanobacteria [117], and Saccharibacteria in biofilms formed on microplastics [118]. In mice exposed to polychlorinated biphenyls (PCBs), among other taxa, the amount of Proteobacteria and Saccharibacteria also increased significantly in the gut microbiota [119].

Concerning the correlations found between TNF-α levels and gut bacterial abundances (Table 2), Saccharibacteria (former designation: TM7) and Proteobacteria were elevated in patients with ulcerative colitis [120]. Moreover, Saccharibacteria are considered to play a role in inflammatory bowel disease [121] and were positively related to pro-inflammatory cytokines in diabetes-induced intestinal inflammation [122]. Multiple factors of Cyanobacteria can also contribute to adverse effects on the intestinal tract and the associated mucosal immune system, including cyanobacterial toxins [123], where exposure to Microcystin-LR, a hepatotoxin produced by freshwater cyanobacteria, resulted in increases in pro-inflammatory transcripts (such as TNF-α) within the colonic tissue of mice [124].

It is of interest to what extent PVC effects in rabbits can be correlated with human health impacts. This comparison may also be justified by the limited number of publications in the literature on PVC effects on humans. Microplastic particles have been detected in various organs in humans [69], as well as in the spleens of rabbits. Weight loss or even weight gain has produced contradictory results in humans, and in this respect, no clear findings could be made in rabbits as well. In sexual prematurity, the results are comparable, as early sexual maturation under microplastic treatment is also seen in humans [32,33]. Inflammatory processes also occur in both humans and rabbits, as indicated by TNF-α studies [85]. Intestinal microbiota changes have also shown similarities in the association of certain taxa with treatments. These were the following taxa: *Odoribacter*, which can cause health problems in humans but can be beneficial in preventing certain diseases [89]; *Alistipes*, which has been shown to have a protective effect for a number of human diseases [91]; and *Rumminococcaceae UCG-005*, which can be associated with human diseases [105]. These results make it plausible that the rabbit, together with the mouse and the rat, could serve as a good model for studying the potential impact of microplastics on human health.

In summary, significant or trend-like changes in the examined rabbit physiological, and pathological parameters were associated with significant or trend-like alterations in the relative abundance of several potentially pathogenic or inflammatory gut bacterial taxa in response to PVC microplastics treatment. Some of these microbiota changes were correlated with blood serum TNF-α levels and the PVC content of the feed. These results, together with the decreasing patterns of the Shannon and Chao1 diversity indices, suggest that the overall changes in the rabbit gut bacterial composition pointed to a gradual shift towards a dysbiotic state of the gut microbiota in response to dietary PVC exposure and are associated with changes in physiological and pathological parameters [125,126].

## 4. Material and Methods

### 4.1. Production of Experimental Feeds

PVC microplastics without any additives were purchased in the size range of 60–250 µm from Werth-Metall GmbH (Erfurt, Germany). The three experimental batches of rabbit diet were produced by INNOVO Ltd. (Isaszeg, Hungary) as follows. Commercially available rabbit feed mixtures (UNI Mother and Fattening Rabbit Feed Mixture, AGRIBRANDS Europe Hungary ZRt., Karcag, Hungary) were milled, mixed, and re-granulated. The feed containing low (P1) and high (P2) doses of PVC consisted of 50 g and 500 g of 60–250 µm PVC microplastics per 100 kg of feed. For control (C), no microplastics were added to the feed, but they underwent the same procedure as those containing microplastics.

### 4.2. Animal and Feeding Experiments

All animal experiments were conducted in accordance with Directive 2010/63/EU and ARRIVE Guidelines. The animal studies, including conditions for animal welfare and handling, were conducted in strict compliance with the recommendations and regulations of the European Animal Research Association [127] and the Science Ethics Code of the Hungarian Academy of Sciences [128]. This study was conducted with the knowledge and approval of the Animal Welfare Body of the Hungarian University of Agriculture and Life Sciences, Szent István Campus. The animal experiment was registered under permission number PE/EA961-7/2020 from the Pest County Governmental Office. All efforts were made to minimise the suffering of the animals involved.

Eighteen female New Zealand White (NZW) rabbits, 12 weeks of age, were provided by Innovo Ltd. (Isaszeg, Hungary) MD stock. The rabbits were divided into three treatment groups, each containing six rabbits. Throughout the experiments, the rabbits were housed under controlled environmental conditions, housed separately in cages, and were given ad libitum access to feed and water. All rabbits were pre-fed with UNI feed for two weeks, followed by 21 days of ad libitum feeding with the experimental diet supplemented according to their respective treatment groups (P1, P2, and C).

### 4.3. Determination of Body and Organ Weights

Animal body weights were measured weekly in all three treatment groups with gram precision. The weekly measurements started at the beginning of the microplastic feeding. The physiological examination of the experimental animals included the weighing of the following organs: spleen, kidney, liver, and ovary at the end of the experiment.

### 4.4. Determination of Hormone and Tumour Necrosis Factor Alpha (TNF-α) Levels

For the TNF-α assay, 100 µL of rabbit serum sample was tested using an ELISA kit (Elabscience CatNo: e-EL-RB0011, Houston, TX, USA) pre-coated with rabbit TNF-α-specific antibody. The assay was performed according to the manufacturer’s instructions. The reactions were then measured photometrically at 450 nm using a Thermo LabSystem Multiskan EX microplate reader (Waltham, MA, USA). The OD value is proportional to the concentration of rabbit TNF-α at pg/mL concentration.

Estradiol (E2) and progesterone (P4) hormone levels were measured similarly to TNF-α, except that 25 µL serum samples were used for E2 measurement with the NovaTec 17-beta-estradiol kit (DNOV003, Immundiagnostica GmbH, Dietzenbach, Germany), and 20 µL of serum sample were used for P4 measurement with the NovaTec Progesterone kit (DNOV006, Immundiagnostica GmbH, Dietzenbach, Germany). Otherwise, the manufacturer’s instructions were followed.

### 4.5. Blood Serum Analyses

The samples were collected from the rabbits’ ear veins into 2 mL Eppendorf tubes. After collecting the whole blood, the sample was left undisturbed at room temperature for 30 min. Subsequently, the clot and serum were separated by centrifugation at 3500 g for 10 min in a refrigerated centrifuge. The centrifugally separated serum was then pipetted into a new Eppendorf tube and stored at −20 °C for further use. The concentration of various parameters including plasma total protein (TP), albumin, AST (U/L), ALT (U/L), ALKP (U/L), α-amylase (U/L), lipase (U/L), glucose (mmol/L), triglycerides (mmol/L), total cholesterol (mmol/L), urea (mmol/L), creatinine (mmol/L), phosphorus (mmol/L), calcium ion (mmol/L), potassium (mmol/L), sodium (mmol/L), and chloride (mmol/L) were determined by the Department of Pathophysiology and Oncology, University of Veterinary Medicine, Budapest, Hungary.

### 4.6. Macroscopic Pathological Examination of Organs

External macroscopic morphological and pathological examinations were performed on the liver, kidneys, spleen, abdominal lymph nodes, caecum, small and large intestine, and rectum of the animals. The findings, recorded during dissection, were subsequently utilised for the assessment of organ lesions in each individual animal.

### 4.7. Histological Examination of the Organs

After the necropsy, the organs were fixed in 4% buffered formaldehyde solution. Histology was performed on the preserved organs of the rabbits (small and large intestine, ileum, caecum, rectum, liver, spleen, kidneys) belonging to the treatment groups P1, P2, and C. The fixed tissues were trimmed, processed, embedded in paraffin, sectioned with microtome, placed on glass microscope slides, stained with hematoxylin and eosin, and examined by light microscopy. The digital photos were taken, and the microscopic evaluation was done with an Olympus microscope (Tokyo, Japan). The magnifications of the Olympus objectives used were as follows: 2×, 10×, 20×, 40×, 100×.

Histological evaluation was carried out by Róbert Glávits, a histopathologist, according to the Hungarian Good Laboratory Practice Regulation: 42/2014 (VIII.19). EMMI decree of the Minister of Human Capacities, which corresponds to the OECD GLP, ENV/MC/CHEM (98) 17.

### 4.8. Metagenomic Sequencing of Rabbit Intestinal Content Samples

For the analysis of the gut microbiota, sections of approximately 10 cm of the ileum, caecum, and rectum were excised and the contents of the intestinal section were transferred into a sterile urine collection vessel. A total of 0.2 g of each sample was stored at −70 °C until use. The collected intestinal content samples were evenly homogenised and gDNA was purified from about 0.1 g of the samples by Xenovea Ltd. (Szeged, Hungary) using the AquaGenomic Kit (MultiTarget Pharmaceuticals, Salt Lake City, UT, USA) following the instructions of the manufacturer. Xenovea Ltd. (Szeged, Hungary) performed 16S rRNA gene-specific amplification, library preparation, and sequencing on the Illumina MiSeq platform (Illumina Inc., San Diego, CA, USA), using 2 × 300 bp paired-end reads with a quality score of Q30 > 75%.

### 4.9. Differential Abundance Testing of the Bacterial Composition of Rabbit Intestinal Content Samples

The bacterial composition of the intestinal microbiota was analysed using the NCBI Nephele QIIME1 pipeline [129]. The sequences were clustered into Operational Taxonomic Units (OTUs) with the open reference method using the reference sequence collection SILVA99 v.132. For clustering to a genus, the similarity threshold was set at 0.94. Low-abundance OTUs were filtered out before differential abundance testing [130,131,132,133] at a relative abundance threshold of ≥0.01% for at least 50% of the examined samples [134,135]. Differential abundance testing between the treatment groups (C, P1, and P2) was performed by the Kruskal–Wallis test using IBM SPSS Statistics 29.0 software (SPSS Inc., Chicago, IL, USA) with *p*-values adjusted by the Bonferroni correction [136,137]. Only *p* < 0.05 values adjusted by the Bonferroni correction were considered statistically significant. The OTU tables were rarefied at the sampling depth of 87,538. The Shannon diversity and Chao1 indices were analysed by One-Way ANOVA followed by the Tukey HSD post hoc test using IBM SPSS 29.0 software. By amplicon sequencing of the 16S V3–V4 variable regions, 95,000–130,000 merged and quality-filtered Illumina reads were obtained per intestinal sample, containing no ambiguous nucleotides. Their mean Illumina Quality Score in the entire 16S V3–V4 amplicon range was >Q30, and thus suitable for further analyses.

### 4.10. General Statistical Analysis

Statistical analyses of the antioxidant parameters, SCFAs, and serum biochemical measurements data were performed using IBM SPSS Statistics 29.0 Win64 Modified Release 1 software (IBM SPSS Inc., Chicago, IL, USA, https://www.ibm.com/spss, released 18 April 2023). The results were subjected to a one-way analysis of variance (ANOVA), and in the case of a significant treatment effect, the Tukey post-hoc test or the Bonferroni correction [136,137] was used to check the differences between the groups. A *p*-value of < 0.05 was considered significant.

## 5. Conclusions

Microplastics are very important environmental pollutants that have received increasing scientific and social attention in recent years. Microplastics can enter the animal’s body in several different ways, but they mainly have their effects when they accumulate in the gut. However, relatively little is known about the toxic effects of microplastics on certain domestic animals, such as rabbits. To the best of our knowledge, this is the first report examining the effects of dietary PVC microplastic exposure on rabbits in a controlled feeding trial involving microbiota analyses as well. Rabbits are exposed to microplastics under farming conditions, including PVC exposure. In our study, we have chosen female rabbits exposed to low and high concentrations of PVC microplastics for 21 days. Compared to untreated controls, intestinal histological changes, pathological abnormalities of organs, changes in reproductive biology, markers of inflammation, the presence of microplastic particles in the spleen, and responses to the gut microbiome were monitored and alterations and changes were demonstrated. A significant and trend-like correlation between the PVC content of the feed and the changes in physiological and pathological parameters and microbiota was identified.

Although there is currently no common standardised regulation on microplastic limits in the European Union [8], our results show that even the low-dose PVC treatment in the P1 group at the proposed limit induced severe changes in rabbits. As a consequence, the PVC-induced lesions highlight the need to pay more attention to the health risks of PVC microplastics to animals and humans. This study may provide new ideas and perspectives on future research about the possible health risks of animals exposed to PVC microplastics.

## Figures and Tables

**Figure 1 ijms-25-12646-f001:**
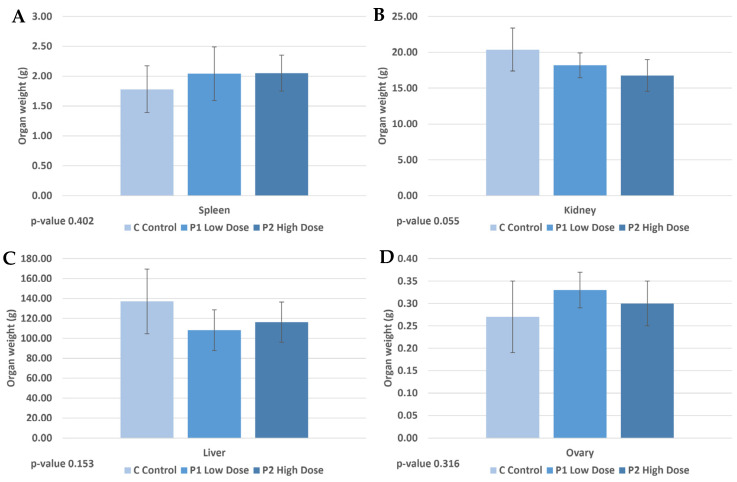
Effect of treatment with P1 (Low Dose) and P2 (High Dose) on organ weights ((**A**) spleen, (**B**) kidney, (**C**) liver, and (**D**) ovary) in comparison to the control C. Each bar represents the mean organ weight (g) with error bars indicating standard deviation (SD). *p*-values are shown below each graph.

**Figure 2 ijms-25-12646-f002:**
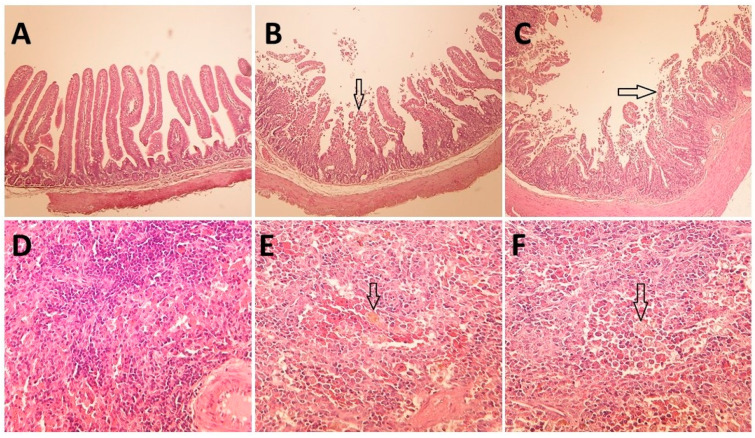
Representative histological samples from the ileum (**A**–**C**) (100× magnification) and spleen (**D**–**F**) (400× magnification), hematoxylin and eosine staining. Tissues of spleen and ileum from the control experiment shown in (**A**,**D**), respectively. (**B**,**C**) indicate intestinal villi abnormalities in the ileum of P1 low dose and P2 high dose experimental group, respectively. (**E**,**F**) show microplastic particles in the spleen in P1 low dose and P2 high dose experimental group, respectively. Arrows indicate particular alterations.

**Figure 3 ijms-25-12646-f003:**
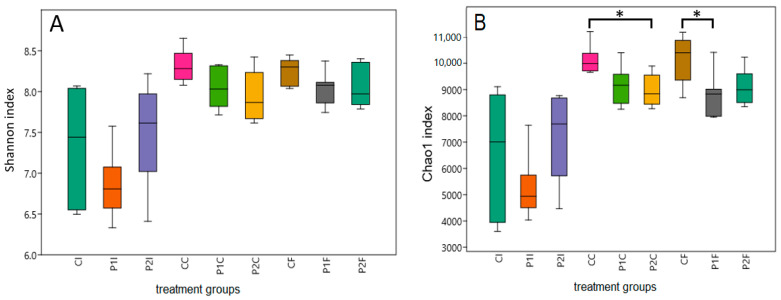
The Shannon (part **A**) and Chao1 (part **B**) diversity index in the different rabbit treatment groups control (C), low PVC (P1), and high PVC (P2), and intestinal sections ileum (I), caecum (C), and faeces (F). * Difference significant at the *p* < 0.05 level.

**Figure 4 ijms-25-12646-f004:**
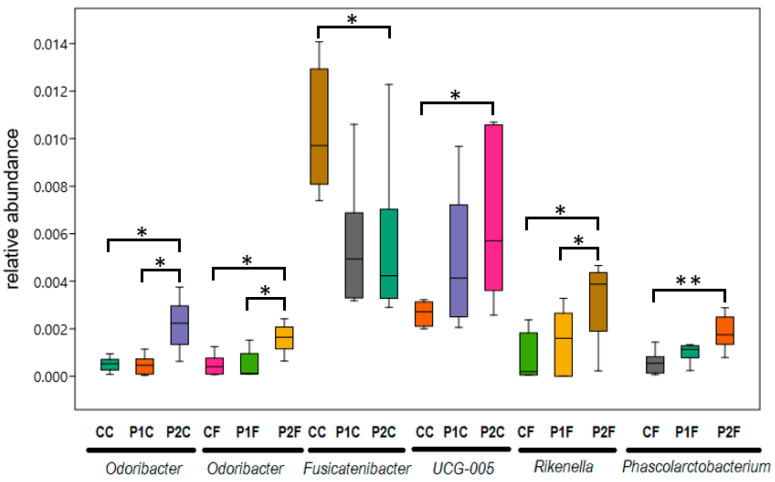
Significant changes at the genus level in relative abundance values between treatment groups control (C), low PVC (P1) and high PVC (P2) within the intestinal sections caecum (C) and faeces (F). The genus *Ruminococcaceae UCG-005* is abbreviated as *UCG-005* in the figure. Relative abundances are displayed on a scale from 0.0 to 1.0. * Difference significant at the *p* < 0.05 level, ** Difference significant at the *p* < 0.01 level. For further details, see Appendix A.

**Figure 5 ijms-25-12646-f005:**
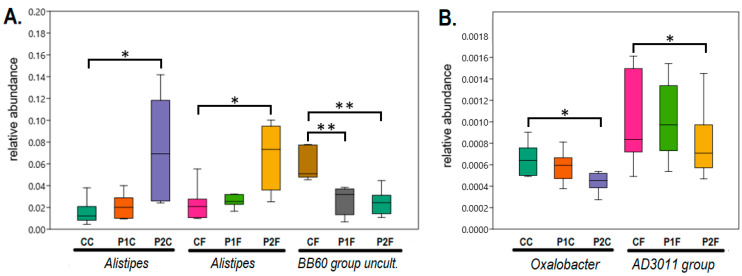
Significant changes at the genus level in relative abundance values between treatment groups control (C), low PVC (P1), and high PVC (P2) within the intestinal sections caecum (C) and faeces (F). For further details, see Appendix A. The abbreviation “*BB60 group uncult*” in (**A**) refers to the *Clostridiales vadin BB60 group*, *uncultured* bacterium, while the abbreviation “*AD3011 group*” in (**B**) refers to the *Clostridiales FamilyXIII AD3011 group*. Relative abundances are displayed on a scale from 0.0 to 1.0. * Difference significant at the *p* < 0.05 level, ** Difference significant at the *p* < 0.01 level.

**Table 1 ijms-25-12646-t001:** Macroscopic alterations of organs in the treatment groups.

Organ	P1 Low PVC Dose	P2 High PVC Dose	C	Kruskal–Wallis Test *p*-Values ^a^	Significant Changes
Swollen liver	3/6	4/6	1/6	*p* = 0.226	none
Abnormal kidneys	0/6	0/6	1/6	*p* = 0.368	none
Swollen Peyer plaque, lymph node	4/5	2/6	6/6	*p* = 0.059	none
Small intestine swollen mucosa	6/6	6/6	2/6	*p* = 0.021	C vs. P1 C vs. P2
Abnormal caecum	2/6	0/6	3/6	*p* = 0.160	none
Abnormal large intestine and rectum	0/6	0/6	0/6	*p* = 1.00	none
Swollen spleen	5/6	5/6	3/6	*p* = 0.351	none

^a^ Significance *p*-values have been adjusted by the Bonferroni correction for multiple tests between treatment groups.

**Table 2 ijms-25-12646-t002:** Pearson correlation coefficients (*R*) between the relative abundances of selected bacterial taxa and between the TNF-α values in blood serum on week 3, or swollen liver, or the PVC microplastic content of experimental rabbit feeds.

Intestinal Section	Pearson Correlation of Bacterial Relative Abundance in the Gut with							
	TNF-α on Week 3			Swollen Liver	The PVC Content of Rabbit Feed			
	Sacchari-bacteria	Cyano-bacteria	UCG-005	Alistipes	Odoribacter	Alistipes	Rikenella	Phascolarcto-bacterium
Ileum						0.562 ^a^		0.566 ^a^
Caecum	0.501 ^a^	0.545 ^a^	0.599 ^a^	0.479 ^a^	0.784 ^b^	0.711 ^b^		
Faeces	0.562 ^a^		0.619 ^b^	0.587 ^b^	0.715 ^b^	0.717 ^b^	0.626 ^b^	0.717 ^b^

UCG-005 is the abbreviation for the genus *Ruminococcaceae UCG-005*. ^a^ Correlations significant at the *p* < 0.05 level. ^b^ Correlations significant at the *p* < 0.01 level.

## Data Availability

The original contributions presented in this study are included in the article/Appendix A, and further inquiries can be directed to the corresponding author/s.

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
