# Peer review of "Effects of Polyvinyl Chloride (PVC) Microplastic Particles on Gut Microbiota Composition and Health Status in Rabbit Livestock"

_ijms, 2024, doi:10.3390/ijms252312646_

Round 1
Reviewer 1 Report
Comments and Suggestions for Authors
Dear Authors, I have carefully read the manuscript " Effects of polyvinyl chloride (PVC) microplastic particles on gut microbiota composition and health status in rabbit live stock ". This is an interesting paper that provides insight into the impact of PVC treatments on physiological, pathological, hormonal, and microbiota changes in female rabbits. The presented results highlight the potential health risks associated with PVC microplastic exposure which are of great importance since PVC pollution has been growing every daz. In my opinion, this an extremely well written manuscript both scientifically and language wise. I couldn't find any flaws.
Minor points:
Comment 1. Authors should highlight health risks associated with PVC microplastic pollution in Introduction section and claim the statements with literature citations.
Comment 2. Authors should provide technical details for light microscope used for Histological examination of the organs e.g. magnifications of the used objectives.

Author Response
Dear reviewer, thank you for your review and our responses are included in the attached file.

Reviewer 2 Report
Comments and Suggestions for Authors
Detailed observations and recommendations are contained in the annex

Language expressions should be more concise and understandable.
Author Response

(The authors gave the same response as above.)

Reviewer 3 Report
Comments and Suggestions for Authors
The manuscript „Effects of polyvinyl chloride (PVC) microplastic particles on gut microbiota composition and health status in rabbit live stock“ by Papp et al. is an interesting and laborious study with many different parameters. The language is readable and well reviewed. However, the paper is not well structured as it represents something between a review and an original research paper. In addition it appears too long. I suggest that the authors focus on the parameters that they investigated and correlate them to influence on human health. The information is there but it has to be streamlined. Also the materials and methods should be placed between the introduction and the results. The authors should work out why PVC is important to be investigated and if the concentrations that they were using represent any environmental case. Even the low dose appears to be high for an environmental case.
The microscopy images, show a type of particle, but how can the authors know that it is their PVC particle, without having it marked or using chemical imaging?
Below are also some other questions:
Can you indicate why rabbits are a good model to study the effects rather than only stating that there are not many studies on this animal.
L26: Why do you consider only 60-250 μm as microplastics? Microplastics are generally <5 mm in diameter
L56: remove the “relatively” in: Apart from a relatively few publications, studies have also been conducted in mammalian farm animals.
L71-73: Do microplastics affect the sex hormone levels or additives in the plastics.
L83-84: There are two “similar” in the sentence. One is sufficient.
L204: The authors write that the values did not change significantly. Is this backed by a calculated p-value?
L229 can you explain what the 16S V3-V4 means for the health status of the rabbits?
L420-422: “In rabbits, low-nicotine tobacco supplementation to animal feed has been shown to increase the abundance of the genus Alistipes along with several other genera [79].” Why is this sentence relevant for this study?
L434-436: “In rabbits, diets supplemented with organic copper resulted in elevated Rikenella and Lachnospiraceae_NK3A20_group levels, which may have contributed to the lower incidence of diarrhea [86].” The same here. Do you want to say that PVC microplastics do indirectly lead to less diarrhea cases in rabibts by favoring these microorganisms? Then it should be shown in the study or differently expressed here.
Author Response

(The authors gave the same response as above.)

Round 2
Reviewer 2 Report
Comments and Suggestions for Authors
Overall
The author has responded to and changed some of the questions I raised earlier. Looking forward to seeing the article published.
进行一些细微的修改。
Reviewer 3 Report
Comments and Suggestions for Authors
The authors have ammended the manuscript as recommended and the manuscript can now be accepted.